# Accumulation and Origin of Phosphorus and Heavy Metals in Citrus Orchard Soils in Jeju Island, South Korea: Potential Ecological Risks and Bioavailability

Tae-Woo Kang [1,*,†] , Hae Jong Yang [1,†], Won-Seok Lee [1], Bon-Jun Koo [2] and Won-Pyo Park [3,*]

[1] Yeongsan River Environment Research Center, National Institute of Environmental Research, Gwangju 61011, Republic of Korea; morning17@korea.kr (H.J.Y.); boystone@korea.kr (W.-S.L.)
[2] Department of Biological Sciences, California Baptist University, Riverside, CA 92504-3297, USA; bonjunkoo@calbaptist.edu
[3] Plant Resources and Environment Major, Faculty of Bioscience and Industry, College of Applied Life Science, Jeju National University, Jeju 63243, Republic of Korea
[*] Correspondence: kangtw@korea.kr (T.-W.K.); oneticket@jejunu.ac.kr (W.-P.P.); Tel.: +82-62-970-3990 (T.-W.K.); +82-64-754-3317 (W.-P.P.)
[†] These authors contributed equally to this work.

**Abstract:** This study investigated the accumulation of total phosphorus (TP) and heavy metals (HMs; Pb, Zn, Cu, Cd, Cr, and Ni) in citrus orchard soils in Jeju Island, Korea, identifying potential soil pollution sources using statistical analysis. Anthropogenic HM pollution was evaluated using the geoaccumulation index and enrichment factors, whereas HM bioavailability was assessed via single extraction. TP, Zn, Cu, and Cr concentrations in citrus orchard topsoil were significantly higher than those in forestland soils, indicating their accumulation in the surface layer. Statistical analyses confirmed that elements with high concentrations were closely related to potential pollution sources accumulated on the surface layer of citrus orchards owing to agricultural activities. Particularly, Zn and Cu accumulation was confirmed to originate from intensive compost and pesticide use in citrus orchards. Cu showed the highest contamination and enrichment of all HMs. However, Zn and Cu fractions, determined via an availability assessment, were dominated by acid or complex compounds, indicating that labile Zn and Cu have potential bioavailability for plants. Nevertheless, their fractions accounted for a small proportion (mean < 15%). Therefore, despite the high pollution levels of Zn and Cu, their availabilities were extremely low, indicating a negligible bioavailability in crops and no impact on aquatic ecosystems.

**Keywords:** citrus orchard; phosphorus; heavy metals; accumulation; origin; single extraction; bioavailability

## 1. Introduction

Agricultural soils play an essential role both in the ecosystem and as the basis for food production. However, they have recently been exposed to various pollutants such as heavy metals (HMs) and phosphorus owing to the rise in industrial development and agricultural activities [1–6]. In particular, the use of fertilizers, for increasing crop production in agricultural soils, and pesticides have generated concerns regarding the accumulation of phosphorus and HMs [7–11]. Accumulation of phosphorus and HMs in agricultural soils has detrimental effects on soil ecosystems, crops, and the quality of water resources, resulting in serious human health problems [12,13].

Phosphorus is an essential nutrient for crop growth and is supplied by commercial fertilizers (e.g., phosphate fertilizers) and livestock manure [11,13,14]. However, because most phosphorus applied to agricultural soils is a non-degradable organic matter or an insoluble complex with metals such as iron and aluminum, plants cannot easily use it, leading to excess fertilization [5,15,16]. Thus, intensive agricultural development and

excessive application of chemical fertilizers and pesticides containing phosphorus have resulted in the global accumulation of this element and losses in agricultural soils [17,18], also leading to the accumulation of HMs. Phosphorus fertilizers are regarded as a potential cause of HM (As, Cr, Zn, Cd, and Cu) accumulation in agricultural soils as they are produced from phosphate rock, which includes several naturally occurring HMs [4,10]. Numerous studies have indicated that HMs are accumulated in agricultural soils through byproducts from livestock farming activities; in particular, Zn and Cu are accumulated through the application of pig manure [1,4,17]. Furthermore, pesticides, used for crop disease control, can also be the origin of HM accumulation [19–21]. For example, various citrus, apple, grape, and avocado orchards have experienced an increase in Cu concentration in the soil when using Cu-based products (e.g., $CuSO_4$, CuO, Bordeaux mixture) [22–27]. HMs accumulated in soil via these agricultural activities exhibit low solubility and are considered potentially toxic, causing various cancers and kidney function disorders in humans when they enter the food chain or through direct contact [2,28–32].

As the components and inputs of fertilizers, livestock manure, and pesticides used differ depending on crop types and soil quality, predicting the accumulation levels of HMs in agricultural soils is difficult [10,33]. Researchers have attempted source assessments through potential environmental risk factors and statistical analysis to determine HM contamination levels [29,34]. Despite such attempts, many researchers have suggested that HM bioavailability assessments via single- or sequential extraction are relatively more appropriate than methods using total concentrations of HMs in agricultural soils [35–37]. Although single-extraction is dependent on several factors, including soil characteristics (e.g., pH and organic matter), crops, and agricultural management systems, it is widely used as a powerful tool to evaluate the HM fractions available to crops in the soil [25,38–40].

Soils in Jeju Island have a high phosphorus retention capacity ($P_{RC}$) and cation exchange capacity (CEC) owing to their main clay minerals, including allophane, ferrihydrite, and Al–humus complexes, resulting in a high fertilizer requirement [41,42]. The amount of chemical fertilizers used per unit area of farmland on the island (460 kg/ha) was the highest in Korea in 2021, excluding metropolitan cities such as Seoul and Busan [43]. In particular, because of the strong phosphorus-fixing ability of active aluminum in allophane and Al–humus complexes, phosphorus fertilizers have been excessively applied [16,44]. The main crop type in Jeju Island is citrus, which accounts for approximately 35% of the total agricultural land [43]. Notably, pesticides have been most frequently applied in citrus orchards to obtain high quality citrus trees with improved yields in Korea [45]. Agricultural activities such as fertilizer and pesticide application to increase citrus fruit production, have led to the accumulation of phosphorus and HMs, possibly contaminating soil and aquatic ecosystems such as rivers [46,47]. Despite existing studies on plant nutrition and residual pesticides in Jeju Island, Korea, there is a lack of research on the accumulation of phosphorus and HMs owing to fertilizer and pesticide use [45,48,49]. Moreover, several studies on HM accumulation in citrus orchards have only targeted the total concentration of HMs, and no bioavailability assessments have considered the impacts of HMs on crops.

Therefore, in this study, we aimed to present the concentration distribution of total phosphorus (TP) and HMs in soil samples from citrus orchards and forestland (control sites) in Jeju Island, South Korea, evaluating their accumulation levels. Furthermore, we aimed to identify the potential pollution sources of TP and HMs based on correlation analysis (CA) and principal component analysis (PCA). In particular, we assessed the anthropogenic pollution levels of HMs using the geoaccumulation index ($I_{geo}$) and enrichment factor (EF) and also identified their bioavailability to plants using a single extraction.

## 2. Materials and Methods

### 2.1. Study Area

Jeju Island (latitude 33°06′–34°00′, longitude 126°08′–126°58′) is located 90 km south of the Korean Peninsula with a total area of 1829 km$^2$, and Halla Mountain, at 1950 m.a.s.l., is in its center (Figure 1). It has a humid subtropical climate with mild winters and relatively

high precipitation. Its annual average temperature range is 15.6–16.9 °C, with an annual average precipitation (1991–2020) that differs depending on each region (Eastern area: 2030 mm; Western area: 1183 mm; Southern area: 1990 mm; Northern area: 1502 mm) [50]. The parent material of the soil is pyroclastic rock originating from basalt, but its features vary greatly depending on soil formation conditions such as climate and vegetation [48,51]. According to the Taxonomical Classification of Korean Soils [52], soils in Jeju Island are classified into 7 orders, 11 suborders, 15 great groups, 28 subgroups, and 66 series, with Andisols accounting for approximately 80% of the total soil types. In particular, Andisols are appropriate for citrus orchards owing to their poor nutrient availability; therefore, Jeju Island has numerous citrus orchards [53].

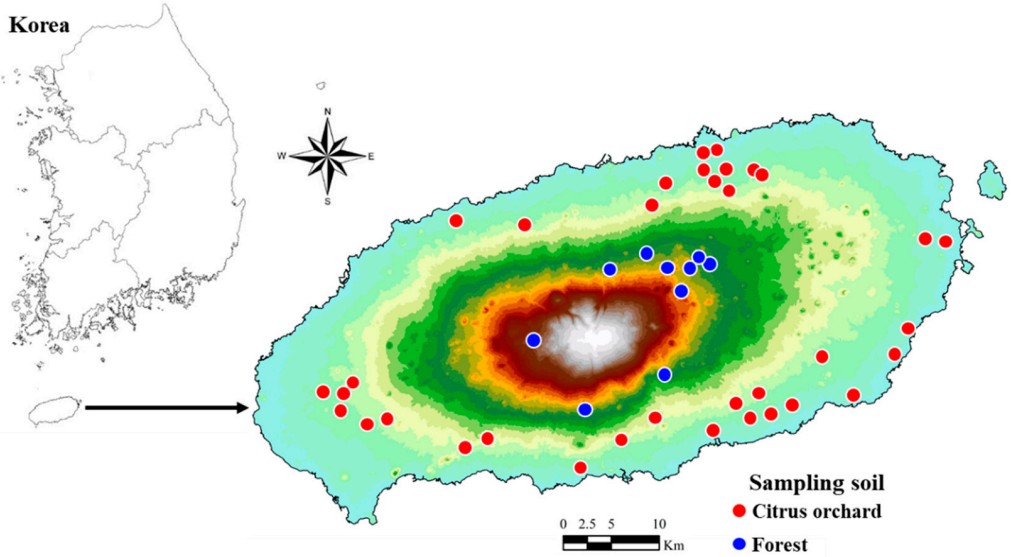

**Figure 1.** Locations of soil samples collected from citrus orchards (red circles) and forests (blue circles) in Jeju Island, Korea.

## 2.2. Sampling and Pretreatment

Topsoil (0–15 cm) and subsoil (15–30 cm) samples were collected from cultivated (35 citrus orchard sites) and uncultivated land (10 forestland sites) (Figure 1). The soil samples from the forestland were used as control sites to assess the accumulation and pollution of TP and HMs in citrus orchards. Five soil samples were collected from each site with a soil auger after removing an organic matter layer of approximately 1 cm and then mixed and homogenized for representativeness of each site [54]. The homogenized soil samples were immediately transferred to the laboratory, air-dried, and passed through a 2 mm sieve for physicochemical and single-extraction analysis. Subsequently, TP and HM analysis were performed via re-passing through a 0.15 mm sieve [55]. A detailed description of the physicochemical analysis is presented in the supplementary materials (Physicochemical analysis). The physicochemical properties of the topsoil and subsoil in citrus orchard and forestland soils are shown in Table S1. Soil organic matter (SOM), soil pH (NaF), $P_{Rc}$, and $Al_o + 1/2Fe_o$ at most sites showed high concentration levels, clearly reflecting the Andic properties of volcanic ash soils.

## 2.3. Phosphorus and Heavy Metal Analysis

The concentrations of TP and six HMs (Pb, Zn, Cu, Cd, Cr, and Ni) were analyzed using aqua regia ($HNO_3$:HCl = 1:3) digestion through the following pretreatment processes (including Al of the Earth's crust) [55]. Air-dried soil samples (<0.15 mm) weighing 0.5 g were placed in a Teflon vessel, and distilled water was added. Additionally, 10 mL of $HNO_3$ (65%) was added to oxidize the organic matter and transferred in a heating block (OD-98-001, ODLAB, Korea). Then, after cooling to approximately 22 °C, aqua regia ($HNO_3$:HCl = 3:9 mL) was added and completely decomposed in a heating block

(at 150–190 °C) for 4 h. The decomposed solution was rinsed with 20 mL of 2% $HNO_3$ and transferred to 100 mL volumetric flasks, after which distilled water was added, and TP and HMs were measured thrice using an inductively coupled plasma optical emission spectrometer (JY138 Ultrace, Jobin Yvon, France).

Quality control for the HM analysis results via aqua regia digestion was evaluated by repeating the analysis three times in the same processes described above with certified reference material (CRM; BAM–U112a, Germany). CRM contains 10 HMs (As, Cd, Co, Cr, Zn, Cu, Hg, Ni, Pb, and V) but not phosphorus; therefore, we only examined the 6 HMs (Cd, Cr, Zn, Cu, Ni, and Pb) analyzed in this study. The analyzed concentrations in CRM are as follows: $3.88 \pm 0.10$ mg/kg (certified value $4.12 \pm 0.15$) for Cd, $78.8 \pm 2.2$ mg/kg (certified value $80.1 \pm 2.5$) for Cr, $185 \pm 4$ mg/kg (certified value $198 \pm 6$) for Zn, $76.5 \pm 3.9$ mg/kg (certified value $75.5 \pm 3.1$) for Cu, $10.9 \pm 0.2$ mg/kg (certified value $10.1 \pm 0.5$) for Ni, and $192 \pm 9$ mg/kg (certified value $198 \pm 8$) for Pb. We considered that the analysis accuracy and precision for these six HMs were satisfactory in the range of 93.6–107.6% and <5% RSD (relative standard deviation), respectively.

*2.4. Pollution Assessment of Heavy Metals*

The anthropogenic pollution assessment of six HMs (Pb, Zn, Cu, Cd, Cr, and Ni) in citrus orchard soils was determined using $I_{geo}$ and EF from the concentrations in the crustal origin and background regions [56,57]. The $I_{geo}$, first suggested by Muller [56], is widely used in soil studies as it enables the evaluation of anthropogenic HM pollution in sediments [58–60]. $I_{geo}$ refers to the relative enrichment level of HMs, which can be obtained by calculating the concentration of each HM and the background concentrations multiplied by the correction constant for crustal origin effects, as shown in Equation (1) [56,57].

$$I_{geo} = \log_2 \left( \frac{C_n}{1.5B_n} \right), \tag{1}$$

where $C_n$ is the concentration of HMs in the topsoil of citrus orchards, $B_n$ is the background concentration of HMs, and 1.5 is the correction constant for crustal origin effects. The background concentrations used the average HM concentrations in the topsoil of uncultivated forestland surveyed in this study: Pb (29.7 mg/kg), Zn (94.1 mg/kg), Cu (29.2 mg/kg), Cd (1.0 mg/kg), Cr (64.5 mg/kg), and Ni (32.4 mg/kg) (Table S2). The calculated $I_{geo}$ values were classified as follows: $I_{geo} < 0$, practically uncontaminated; $0 \leq I_{geo} < 1$, uncontaminated to moderately contaminated; $1 \leq I_{geo} < 2$, moderately contaminated; $2 \leq I_{geo} < 3$, moderately to heavily contaminated; $3 \leq I_{geo} < 4$, heavily contaminated; $4 \leq I_{geo} < 5$, heavily to extremely contaminated; and $5 \leq I_{geo}$, extremely contaminated.

EF is used as a parameter to evaluate the enrichment level of each HM from changes in the relative content of crustal origin elements. Here, aluminum (Al) from the Earth's crust was used, and the EF was calculated to exclude particle size effects with Equation (2) [57].

$$EF = \left( \frac{(M/Al)_{sample}}{(M/Al)_{background}} \right), \tag{2}$$

where $(M/Al)_{sample}$ refers to the ratio of HM and Al concentrations in the topsoil of citrus orchards, and $(M/Al)_{background}$ indicates the ratio of HM and Al concentrations in the background samples (topsoil in uncultivated forestland). The calculated EF values were classified as follows: $EF < 2$, deficiency to minimal enrichment; $2 \leq EF < 5$, moderate enrichment; $5 \leq EF < 20$, significant enrichment; $20 \leq EF < 40$, very high enrichment; and $40 \leq EF$, extremely high enrichment.

$I_{geo}$ and EF values were assessed based on the suggested pollution criteria, which are frequently used to indicate anthropogenic pollution in the soil environment [29,34,58–60].

### 2.5. Heavy Metal Bioavailability Using Single Extraction

Bioavailability using single extraction is a widely used index to evaluate HM availability in crops [31,61,62]. The bioavailabilities of five HMs (Pb, Zn, Cu, Cr, and Ni) in citrus orchard topsoils were determined using acid extractants (hydrochloric acid, HCl), complexing agents (diethylenetriamine pentaacetic acid–triethanolamine, DTPA–TEA; ethylene diamine tetra acetic acid, EDTA), and neutral salt extractants (ammonium acetate, $NH_4OAc$) [31,62–65]. For HCl extraction, 10 g of air-dried soil (<2 mm) was placed in a 100 mL Erlenmeyer flask, adding 50 mL of 0.1 M HCl solution, after which it was shaken at 30 °C and 120 rpm for 1.5 h [31,63]. For DTPA–TEA extraction, 10 g of air-dried soil (<2 mm) were placed in a 100 mL polyethylene Erlenmeyer flask with 20 mL of 0.005 M DTPA–TEA solution (pH = 7.3) and then shaken at 25 °C and 120 rpm for 2 h [31,65]. For EDTA extraction, 10 g of air-dried soil (<2 mm) was placed in a 100 mL polyethylene Erlenmeyer flask; subsequently, 50 mL of 0.05 M EDTA solution (pH = 7.0) was added, and the flask was shaken at 25 °C and 120 rpm for 1 h [31,64]. Regarding $NH_4OAc$ extraction, 10 g of air-dried soil (<2 mm) was placed in a 100 mL Erlenmeyer flask with 50 mL of 1.0 M $NH_4OAc$ solution (pH = 7.0); the mixture was shaken at 30 °C and 120 rpm for 1.5 h [31,66]. Thereafter, all extraction solutions were filtered and measured with an inductively coupled plasma optical emission spectrometer (JY138 Ultrace, Jobin Yvon, France) three times.

### 2.6. Statistical Analysis

Statistical analysis was conducted via CA and PCA using the Statistical Package for Social Sciences, ver. 21 for Windows, to evaluate the pollution sources of TP and HMs in citrus orchard soils. CA is a method used to measure the relationship between two groups. The evaluation was performed using Pearson's correlation coefficient (*r*) and significance level (*p*), which are most commonly used in parametric statistics. Pearson's correlation coefficient can be calculated through standardization of covariance; here, covariance is the variance between two variables, and standardization is obtained by dividing the covariance into standard deviations of two variables [67,68]. PCA, which was first proposed by Pearson [69], is one of the most widely used multivariate statistical methods and has been used to evaluate the pollution source origin of HMs in the soil environment [3,29,34,69]. We only considered the extracted factors with eigenvalues $\geq$ 1.0 [19,70]. The data were orthogonally rotated using the commonly used varimax method for a clearer interpretation [71,72].

## 3. Results and Discussion

### 3.1. Total Concentrations of Phosphorus and Heavy Metals

The concentration distributions of TP and six HMs (Pb, Zn, Cu, Cd, Cr, and Ni) in soils (topsoil and subsoil) collected from citrus orchards and forestlands are shown as a box-and-whisker plot (including Al from the Earth's crust) (Figure 2). The concentrations of TP, Zn, Cu, Cr, and Ni in citrus orchard soils were higher than those in forestland soils (control sites), whereas those of Pb, Cd, and Al in citrus orchard soils were not significantly different from those in forestland soils. Phosphorus and HMs in forestland topsoil and subsoil showed similar concentration levels, implying that they accurately reflected the characteristics of control sites for the contamination evaluation of citrus orchard soils. However, the Pb concentration in forestland topsoil tended to be slightly higher than that in forestland subsoil as well as that in citrus orchard topsoil and subsoil, but no significant differences were observed. TP and three HMs (Zn, Cu, and Cr) that showed higher concentrations in citrus orchards than in forestlands, showed higher concentrations in the topsoil than in the subsoil, suggesting that they were accumulated on the surface layer of citrus orchards. Such characteristics are readily discernible through the examination of concentration ratios of TP and HMs in the topsoil and subsoil of citrus orchards and forestland (Table S2). The concentration ratios of TP, Zn, Cu, and Cr were higher in citrus orchards than in the forestland, with values of 2.5 (TP) > 1.7 (Cu) > 1.4 (Zn) $\approx$ 1.3 (Cr). This implies that these elements were accumulated on the surface layer of citrus orchards, particularly phosphorus, with the highest accumulation. Concentration ratios higher than 1.0 have been attributed

to agricultural activities in citrus orchards [73,74]; the highest amounts of fertilizers and pesticides in Korea are reportedly applied in the agricultural fields of Jeju Island [43]. In particular, phosphorus reportedly accumulates due to insolubilization of active aluminum in soils containing allophane and Al–humus complexes as well as to long-term excessive application of phosphorus fertilizer [5,15,16].

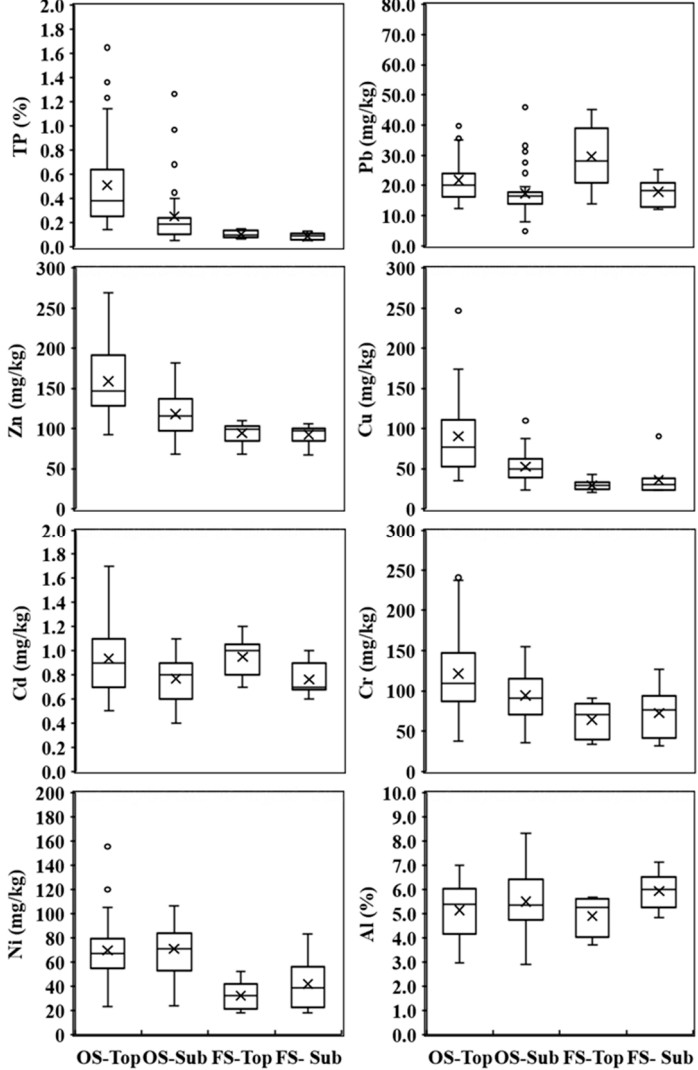

**Figure 2.** Box-and-whisker plots showing the distribution of total phosphorus (TP) and heavy metals (Pb, Zn, Cu, Cd, Cr, and Ni) in topsoil (Top) and subsoil (Sub) collected from citrus orchards (OS) and forestland (FS) in Jeju Island, South Korea (including Al from the Earth's crust). Box-and-whisker plots indicate the median (horizontal line inside the box), mean ("×" in the box), the interquartile range (IQR, rectangular box), 1.5 IQR (a straight line outside the box), and outlier (points outside 1.5 IQR).

For spatial distribution (Figure 3), the concentrations of TP and HMs in the citrus orchard topsoil showed various distribution patterns, with concentrations found to be high in citrus orchards in the northwest and southwest (except for Al from the earth's crust). In particular, the concentrations of TP and three HMs (Zn, Cu, and Cr) that are assumed to accumulate on the surface layer of citrus orchards were observed to be high in numerous citrus orchards, whereas the concentrations of Pb, Cd, and Ni were found to be high in only some citrus orchards. These tendencies may be attributed to the years of cultivation in each citrus orchard. As fertilizers and pesticides have been applied in citrus orchards for a long time, with a different cultivation management system than that of other arable land,

nutrients and HMs are very likely to accumulate on the surface layer as the cultivation years increase. Previous studies on soils in Chinese apple, grape, and cherry orchards and greenhouses (including general farmland) have also found a higher accumulation of HMs and macronutrients as the cultivation years increased [25,28,75].

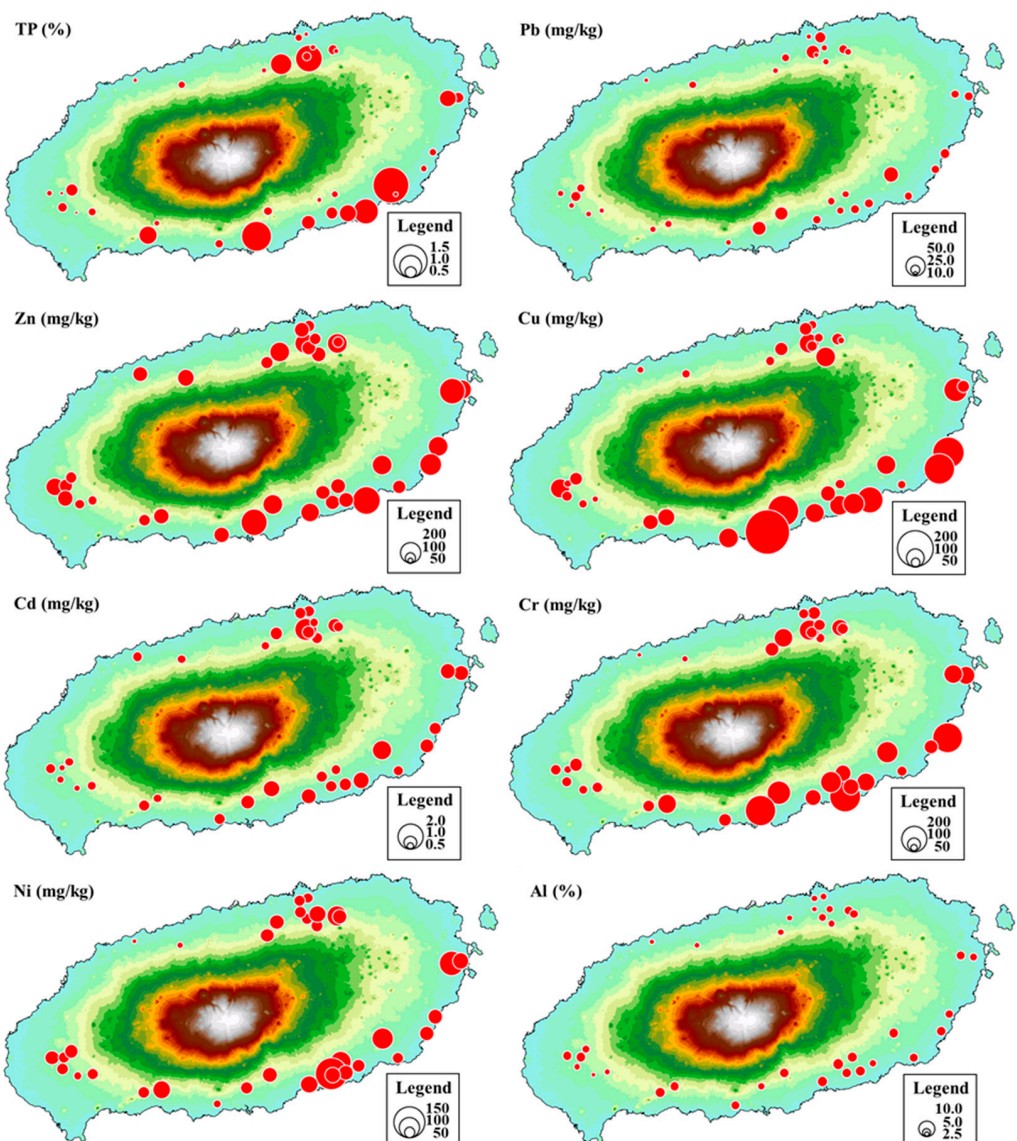

**Figure 3.** Spatial distributions of TP and heavy metals (Pb, Zn, Cu, Cd, Cr, and Ni) in topsoil collected from citrus orchards in Jeju Island, South Korea (including Al from the Earth's crust).

In addition, the concentrations of TP and HMs in citrus orchard topsoil, except for Al, were compared with those in soils in Korea (including Jeju Island) and in other countries with characteristics of volcanic ash origin (Table 1). The average concentrations of TP, Pb, Zn, Cu, Cd, Cr, and Ni in citrus orchard topsoil obtained in this study were 5117 ± 3667 mg/kg, 21.8 ± 6.9 mg/kg, 159 ± 44 mg/kg, 90.5 ± 47.8 mg/kg, 0.94 ± 0.27 mg/kg, 122 ± 51 mg/kg, and 69.5 ± 25.2 mg/kg, respectively, which are higher than those in agricultural soils in Jeonnam, Korea. Simultaneously, the average concentrations of TP, Pb, Zn, Cu, and Cr obtained were comparable to those in agricultural soils in Jeju Island; Cd and Ni average concentrations showed different tendencies. Compared to the agricultural soils of other countries, the average concentration of TP obtained for the citrus orchard topsoil of Jeju Island was higher. Similarly, the average concentration of HMs was slightly lower than or similar to that of the citrus orchard topsoil in Jeju Island. Contrastingly, compared to the

concentrations obtained in this study, the concentration of Ni was higher in Greece, those of Pb and Cu were higher in the Azores archipelago, and those of Cr and Ni were higher in Ecuador. TP and HMs, which showed high concentration levels in the agricultural soils of Korea and other countries, have been reported to accumulate owing to the long-term application of fertilizers and pesticides. Meanwhile, except for that of Pb, the concentrations of HMs in citrus orchard topsoil found in this study were higher than both the background concentration in Korea and the global average concentration. Nevertheless, as these HM concentrations do not exceed the Korean warning criteria guideline [76], a serious level of contamination was not observed.

**Table 1.** Comparison of total phosphorus and heavy metal concentrations (mg/kg) in topsoil collected from citrus orchards in Jeju Island, South Korea, and volcanic ash-derived soils in other countries.

| Country | Region | TP [a] | Pb | Zn | Cu | Cd | Cr | Ni | Reference |
|---|---|---|---|---|---|---|---|---|---|
| Korea | Jeonnam | - [b] | 13.5 [c] (5.0–95.1) [d] | 86.8 (17.8–275) | 21.1 (2.8–106) | 0.31 (0.06–0.69) | 29.3 (2.6–139) | 13.8 (1.3–43.2) | [77] |
| | Jeju Island | 14,000 [e] (9000–25,000) | 45 (32–72) | 168 (136–198) | 62 (46–98) | 0.3 (0.2–0.6) | 222 (155–352) | 142 (110–186) | [78] |
| | Jeju Island | 5117 ± 3667 (1441–16,500) | 21.8 ± 6.9 (12.5–39.8) | 159 ± 44 (92.0–269) | 90.5 ± 47.8 (35.2–247) | 0.94 ± 0.27 (0.50–1.70) | 122 ± 51 (37.4–245) | 69.5 ± 25.2 (23.1–156) | This study |
| Greece | Peloponnese, Argolida | 1270 ± 1500 (180–15,150) | 19.7 ± 7.4 (3.17–48.5) | 74.9 ± 32.8 (23–288) | 74.7 ± 63.9 (11.9–653) | 0.54 ± 0.69 (0.07–6.1) | 83.1 ± 48.2 (28.1–354) | 147 ± 120 (43.8–1258) | [3] |
| Italy | Solofrana | - | - (21–98) | - (92–135) | - (77–565) | - | - (137–335) | - (56–84) | [79] |
| Azores archipelago | São Miguel Island | 1816 | 40.0 | 214 | 156 | 0.39 | 37.4 | 58.6 | [80] |
| Japan | All | 1500 ± 1300 (87–11,000) | 24 ± 50 (1.0–1100) | 89 ± 42 (2.5–330) | 48 ± 48 (0.88–230) | 0.33 ± 0.28 (0.021–3.4) | 58 ± 38 (1.4–230) | 26 ± 21 (0.20–110) | [74] |
| Ecuador | Galápagos Island | - | 5.92 (0.855–11.6) | 190 (64.7–430) | 112 (26.3–170) | 1.17 (0.078–4.11) | 219 (45.3–648) | 152 (15.5–524) | [81] |
| Korean background level | | - | 18.43 | 54.27 | 15.26 | 0.29 | 222.6 | 17.68 | [82] |
| Korean warning criteria guideline | | - | 200 | 300 | 150 | 4 | 200 | 100 | [76] |
| World soil average | | - | 27 | 70 | 38.9 | 0.41 | 59.5 | 29 | [83] |

Note: [a] Total phosphorus, [b] no data, [c] mean or mean ± SD, [d] range (min–max), [e] presented as $P_2O_5$.

### 3.2. Source Origin of Phosphorus and Heavy Metals

CA and PCA were used to identify the potential pollution sources of TP and HM accumulation in citrus orchard topsoil (Figure 4). According to the CA result (Figure 4a), TP, Pb, Zn, Cu, Cd, and Cr in citrus orchard topsoil showed strong positive correlations ($r = 0.50$–$0.74$, $p < 0.01$) among all items, and Cr–Ni also showed a strong positive correlation ($r = 0.47$, $p < 0.01$). Contrastingly, Al showed a weak positive correlation with Cr ($r = 0.35$, $p < 0.05$) but a strong positive correlation with Ni ($r = 0.61$, $p < 0.01$). Further, TP and several HMs showed weak and strong positive correlations with SOM (excluding Pb and Ni; $r = 0.39$–$0.64$) but negative correlations with CEC (excluding Cu, Ni, and Al; $r = 0.35$–$0.63$). These results clearly indicate that the accumulation of TP and HMs in soils with volcanic ash origin is strongly associated with soil physicochemical properties. As above, phosphorus and several HMs with high correlations shared common sources and mutual dependences, showing identical behavior, which implies that potential pollution source interpretation is possible [30,84]. Regarding the principal components derived from PCA, we found two factors (PC1 and PC2) with an eigenvalue >1.0 (Table S3, Figure 4b). As the cumulative variance of these two factors was 71.8% of the total, it was possible to interpret the analysis results. The total variances of PC1 and PC2 were 53.3% and 18.5%, respectively. PC1 showed a high correlation with TP, Pb, Zn, Cu, Cd, and Cr, whereas PC2 showed a high correlation with Ni and Al.

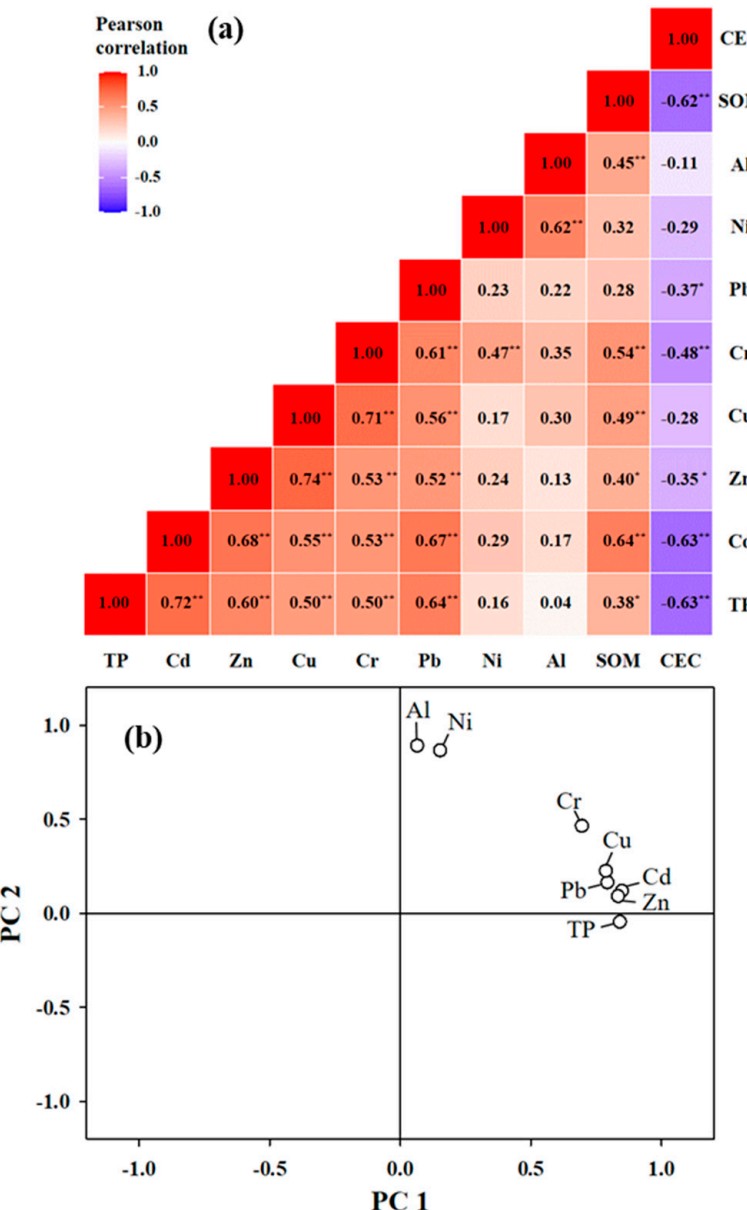

**Figure 4.** Relationships between phosphorus and heavy metals in topsoils collected from citrus orchards in Jeju Island, South Korea, through (**a**) correlation analysis (CA) and (**b**) principal component analysis (PCA) biplot. * Significant correlation at the 0.05 level (one-tailed), ** Significant correlation at the 0.01 level (two-tailed). PC1—principal component 1, PC2—principal component 2.

Based on the CA and PCA results, the pollution sources of phosphorus and HMs in the citrus orchard soils in Jeju Island were clearly divided into two groups, with items strongly associated with anthropogenic (TP, Pb, Zn, Cu, Cd, and Cr) or natural (Ni and Al) factors. Several previous studies have reported that Pb, Zn, Cu, and Cd are derived from anthropogenic factors (pesticides, fertilizers, livestock manure), with higher concentrations of these HMs in arable and agricultural soils than those in background soils [4,85–87]. TP has also been reported to accumulate in soils owing to agricultural activities such as excessive application of phosphate fertilizers [5,6,8]. Our findings are consistent with those of other studies on citrus and apple orchards with similar cultivation management systems to those applied in this study area [9,88–90].

Cd is closely related to phosphate fertilizers along with TP [2,7,11,91], and phosphate fertilizers are known to contain high concentrations of Cd [10,92,93]. Furthermore, the accumulation of phosphorus in soil is associated with intensive livestock production [11,14].

Regarding Pb, gases from industrial complexes and vehicles are deposited on the soil through atmospheric deposition; however, in agricultural soils, mineral fertilizers are the main contributor [3]. Zn is known to originate from anthropogenic or natural sources in agricultural soils [3,27,94], but the application of pesticides, mineral fertilizers, and livestock manure is responsible for its high concentration in soils [1,3,4,80]. Notably, numerous studies have indicated that the application of phosphate fertilizer plays a crucial role as an anthropogenic source of Zn [33]. Cu is a representative HM, generally generated via agricultural activities; the usage of commercial fertilizers and Cu-based pesticides and fungicides is an important contributor to higher concentrations of Cu in the soil [23,27,95]. The application of Cu-rich products (e.g., $CuSO_4$, CuO, Bordeaux mixture) to control fungal pathologies (e.g., Botrytis, *Plasmopara viticola*) in vine cultivation in Italy reportedly results in soil pollution [22]. Cu accumulates in citrus and ginseng cultivation in Korea due to the application of lime Bordeaux mixture [24,26].

In contrast, Cr and Ni are known to be influenced by natural sources because they depend on the parent rock [2,70,96,97]. Al is also a main component of the earth's crust and has been mainly used as a reference element [98,99]. Nevertheless, the presence of Cr was found to be related to the use of nitrogen, phosphorus, and potassium fertilizers in agricultural soils [7,100], and long-term inputs of sludge, compost, and pesticides have been reported as major sources [74,78,79]. Therefore, Cr was found to be influenced by both anthropogenic and natural factors, with the former being more dominant, which was already well-reflected by its high concentration ratio in the topsoil and subsoil.

Our results indicate that TP and three HMs (Zn, Cu, and Cr) with high concentrations in citrus orchard topsoil were strongly associated with potential pollution sources that are commonly used, such as phosphate fertilizers, chemical fertilizers, compost, and pesticides. These elements accumulated on the surface layer of citrus orchard soil owing to agricultural activities. In particular, the accumulation of Cu and Zn, as suggested by Lim et al. [26], was attributed to the application of compost, lime Bordeaux mixture, oxine-copper, cuprous oxide, dithianon, copper hydroxide, copper sulfate, and Zineb (containing high concentrations of Zn and Cu), which are frequently used in citrus orchards.

*3.3. Pollution Assessment of Heavy Metals*

The anthropogenic pollution distributions for six HMs (Pb, Zn, Cu, Cd, Cr, and Ni) in citrus orchard topsoil assessed using $I_{geo}$ and EF are shown as a box-and-whisker plot, and the distribution of $I_{geo}$ and EF pollution classes for each HM is also presented (Figure 5). The pollution class results for each HM are expressed as a percentage of the total sites. The median and mean $I_{geo}$ values for six HMs suggested that there was no contamination. However, the distribution of the pollution class of $I_{geo}$ indicated that Cr and Ni showed a moderate contamination level (Cr, n = 3; Ni, n = 4) in some sites. Cu also showed a moderate level (n = 14) and moderate-to-heavy level (n = 1) of contamination at numerous sites. Similar to $I_{geo}$, the median and mean values of EF indicated no enrichment effect for four HMs, but moderate levels of enrichment of Cu and Ni were found. The distribution of EF pollution class showed a moderate enrichment level for four HMs: Zn, n = 7; Cd, n = 1; Cr, n = 10; and Ni, n = 18. In particular, Cu showed a moderate level (n = 22) and significant level (n = 4) of enrichment at numerous sites, similar to $I_{geo}$, suggesting an effect from anthropogenic pollution sources.

Overall, citrus orchard topsoil was found to have insignificant contamination levels from Pb, Zn, Cd, and Cr, whereas Cu and Ni raised contamination concerns. Although no serious concern is associated, Cu and Ni as well as Zn, Cd, and Cr were found to be affected by anthropogenic sources at several sites. Unlike the findings of previous sections, Ni, which was observed to have a high contamination level, showed a higher concentration level in citrus orchards than in forestlands (control sites). Nevertheless, the concentration ratios of Ni in topsoil and subsoil (Table S2) were similar to each other, and as they were mostly <1.0, the natural effect was more dominant than the accumulation on the surface layer. These results exhibit strong correlations with the soil characteristics of Jeju Island,

which has basalt as its parent material [26]. Additionally, they were consistent with the findings of another study, which indicated the combined effect of high concentrations owing to the parent rock (basalt) and the tendency to accumulate in highly weathered soils [80]. For its part, a moderate enrichment level of Cd was confirmed at one site among all citrus orchard sites, which was considered negligible. Therefore, among the six HMs, Zn, Cu, and Cr were found to be affected by anthropogenic sources in citrus orchard topsoil, with Cu showing relatively higher contamination levels and enrichment than other HMs.

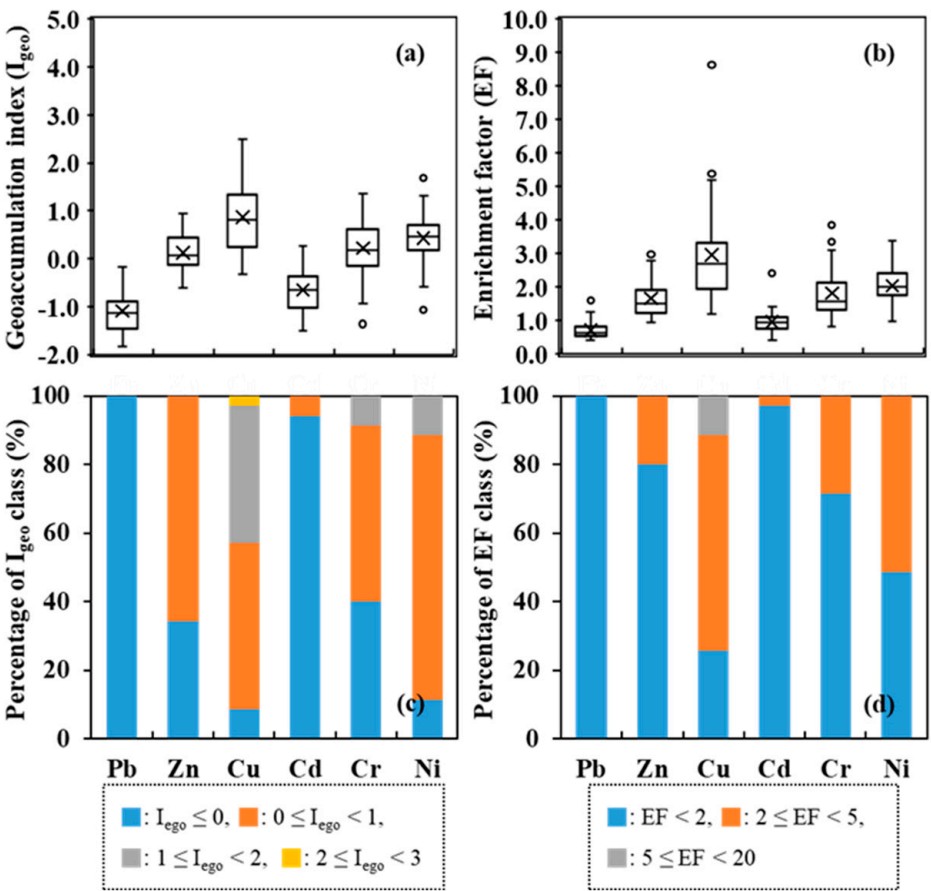

**Figure 5.** Box-and-whisker plots showing the distribution of (**a**) geoaccumulation index ($I_{geo}$) and (**b**) enrichment factor (EF) in topsoil collected from citrus orchards in Jeju Island, South Korea, and the percentage of (**c**) $I_{geo}$ and (**d**) EF pollution class. Box-and-whisker plots indicate the median (horizontal line inside the box), mean ("×" in the box), the IQR (rectangular box), 1.5 IQR (a straight line outside the box), and outlier (points outside 1.5 IQR). $I_{geo} < 0$: practically uncontaminated, $0 \leq I_{geo} < 1$: uncontaminated to moderately contaminated, $1 \leq I_{geo} < 2$: moderately contaminated, $2 \leq I_{geo} < 3$: moderately to heavily contaminated. EF < 2: deficiency to minimal enrichment, $2 \leq EF < 5$: moderate enrichment, $5 \leq EF < 20$: significant enrichment.

### 3.4. Bioavailability of Heavy Metals

To evaluate the bioavailability of HMs in citrus orchard topsoil, the distribution of five available HM (Pb, Zn, Cu, Cr, and Ni) fractions extracted via four single extractions (0.1 M HCl, 0.005 M DTPA–TEA, 0.05 M EDTA, and 1.0 M NH$_4$OAc) are graphically presented in a box-and-whisker plot (Figure 6). The fractions are expressed in percentages, that is, the concentration ratio obtained in each single extraction for each total concentration.

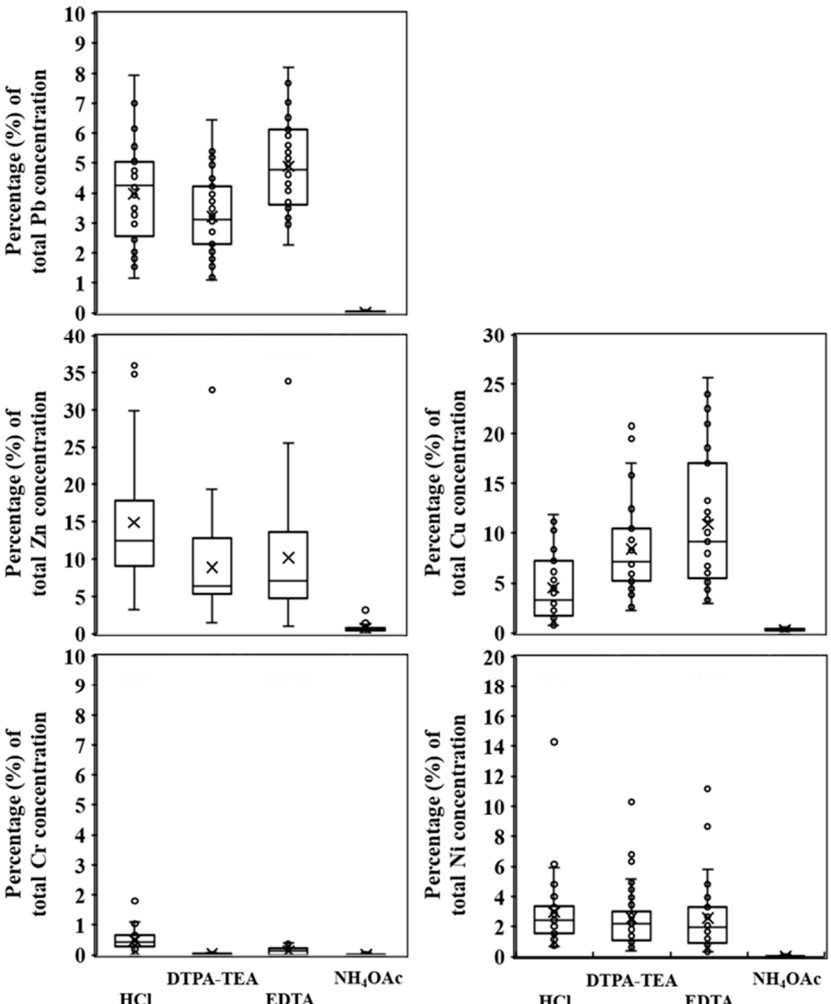

**Figure 6.** Box-and-whisker plots showing the fractionation of heavy metals via single extraction (0.1 M HCl, 0.005 M DTPA-TEA, 0.05 M EDTA, 1.0 M NH$_4$OAc) in topsoil collected from citrus orchards in Jeju Island, South Korea. Box-and-whisker plots indicate the median (horizontal line inside the box), mean ("×" in the box), the IQR (rectangular box), 1.5 IQR (a straight line outside the box), and outlier (points outside 1.5 IQR).

Overall, the levels of the five available HM fractions extracted via four single extractions were mostly <15%, and the fractions were notably rarely extracted using NH$_4$OAc. Looking at each HM, the fractions of available Zn and Cu, which were extracted using HCl, EDTA, and DTPA–TEA single extraction, were higher than those of other HMs, and they were extracted up to approximately 30% at several sites (except for available Cu using HCl). In contrast, the fractions of all HMs extracted using NH$_4$OAc ranged between 0.02% and 0.69%, with almost no available HMs extracted. These findings showed correspondingly higher and lower fractions of Zn (6%) and Cu (24%) derived using EDTA than those obtained in a study on citrus orchards with Mediterranean calcareous soils in Greece [101]. Our results were comparable to those of a different study that obtained fractions of HMs (Pb, Zn, Cu, Cr, Ni, Fe, and Mn) derived using DTPA single extraction for volcanic agricultural soils in southwestern Italy [78]. In addition, the results of the present study were consistent with those of previous research studies evaluating the main agricultural land of the Tongling mining area in China, where the concentrations of HMs (Pb, Zn, Cu, Cd, Fe, and Mn) extracted using NH$_4$OAc single-extraction were lower than those obtained using other extractants (DTPA, EDTA, NH$_4$NO$_3$, and HCl) [31,84]. However, our results were all significantly low compared to the five HM (Pb, Zn, Cu, Cr, and Ni) fractions obtained in

orchard topsoil in Korea (apple, pear, and grape) extracted using three extractants (HCl, DTPA, and EDTA) [102].

In general, although bioavailability using single extraction is widely used to evaluate the availability of HMs in crops in agricultural soils [31,61,62,64,84,101], results can differ depending on the soil physicochemical properties (pH, organic matter, and CEC), types of pollution sources, and agronomic management practices [31,80,84,101]. Therefore, in this study, the bioavailability of HMs in citrus orchard topsoil is assumed to be closely related to the soil properties of allophane and Al–humus complex, whose parent material is pyroclastic rock that originated from basalt [51,78].

These results indicate that HMs in the citrus orchard topsoil in Jeju Island were dominated by acid or complex compounds rather than exchangeability. In particular, among the HMs (Zn, Cu, and Cr) whose concentrations were detected as high due to anthropogenic factors in the previous sections, the acid or complex compound fraction of Zn and Cu was the highest, whereas that of Cr was the lowest. These results indicate that labile Zn and Cu in citrus orchard topsoil have potential bioavailability for uptake from plants. Nevertheless, their acid or complex compound fraction accounted for only a very small proportion (mean < 15%). Although the pollution level was high, the concentrations of available Zn and Cu were extremely low, thereby their bioavailability for crops was assumed to be negligible.

## 4. Conclusions

This study investigated the accumulation and potential pollution sources of phosphorus and HMs in citrus orchard soil and evaluated the anthropogenic pollution level and bioavailability of HMs. The concentrations of TP, Zn, Cu, and Cr in citrus orchard topsoil were significantly higher than those in forestland soils (control sites) as well as in subsoil, and the ratios of their concentrations to topsoil and subsoil in citrus orchard soils were all determined to be >1. This implies their accumulation on the surface layer of citrus orchards; in particular, the accumulation of phosphorus was the highest. Additionally, we found that TP, Pb, Zn, Cu, Cd, and Cr in citrus orchard topsoil had strong positive correlations among all items, and Cr–Ni also showed a strong positive correlation. Contrastingly, Al had weak and strong positive correlations with Cr and Ni. Through PCA, we found two principal components—PC1 (TP, Pb, Zn, Cu, Cd, and Cr) and PC2 (Ni and Al)—with high correlations between the items in each component. Based on these results, phosphorus and HM sources were clearly divided into two groups: anthropogenic and natural factors. Thus, high concentrations of TP, Zn, Cu, and Cr were closely related to potential pollution sources (chemical fertilizers, compost, and pesticides), which accumulated on the surface layer owing to agricultural activities, particularly compost and pesticides related to Zn and Cu accumulation. Furthermore, the $I_{geo}$ and EF assessments suggested that the concentrations of Zn, Cu, and Cr in citrus orchard topsoil were affected by anthropogenic sources; in particular, Cu showed relatively higher contamination levels and enrichment than exhibited by other HMs.

However, most fractions of HMs obtained via four single extractions in citrus orchard topsoils were <15% and were dominated by acid or complex compounds rather than exchangeability; fractions via $HH_4OAc$ were rarely extracted. The acid or complex compound fraction of Zn and Cu was the highest, whereas that of Cr was the lowest. These results indicate that although the pollution level of Zn and Cu was high, the available concentration was extremely low, thereby the bioavailability for crops was assumed to be negligible. Nevertheless, the application of chemical fertilizers, compost, and pesticides should be appropriately controlled, as they result in TP and some HM accumulation on the surface layer of citrus orchards. In particular, it is necessary to minimize the excessive application of phosphate fertilizers by preparing measures to effectively use plants to reduce phosphorus accumulation.

**Supplementary Materials:** The following supporting information can be downloaded at: https://www.mdpi.com/article/10.3390/w15223951/s1, Table S1: Summary of physicochemical properties in the topsoil and subsoil samples collected from citrus orchard and forestland sites in Jeju Island, South Korea; Table S2: Concentration of total phosphorus (TP) and heavy metals in the topsoil and subsoil collected from citrus orchard and forestland sites in Jeju Island, South Korea; Table S3: Principal component analysis (PCA) results of phosphorus and heavy metals in the topsoil collected from citrus orchard sites in Jeju Island, South Korea. References [54,103] are cited in Supplementary Materials file.

**Author Contributions:** Conceptualization, T.-W.K., H.J.Y., W.-S.L., B.-J.K. and W.-P.P.; methodology, T.-W.K. and W.-P.P.; validation, T.-W.K. and W.-P.P.; formal analysis, T.-W.K. and W.-P.P.; investigation, T.-W.K. and W.-P.P.; data curation, T.-W.K., H.J.Y. and W.-P.P.; writing—original draft preparation, T.-W.K., H.J.Y. and W.-P.P.; writing—review and editing, T.-W.K., W.-S.L., B.-J.K. and W.-P.P.; visualization, T.-W.K. and H.J.Y., T.-W.K. and H.J.Y. contributed equally. All authors have read and agreed to the published version of the manuscript.

**Funding:** This study was partially supported by a grant from the National Institute of Environmental Research (NIER), funded by the Ministry of Environment (MOE) of the Republic of Korea (NIER-2021–01–01–129).

**Data Availability Statement:** The data presented in this study are available on request from the corresponding author.

**Conflicts of Interest:** The authors declare no conflict of interest.

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
