# Peer review of "Accumulation and Origin of Phosphorus and Heavy Metals in Citrus Orchard Soils in Jeju Island, South Korea: Potential Ecological Risks and Bioavailability"

_water, doi:10.3390/w15223951_

Round 1

Reviewer 1 Report

Comments and Suggestions for Authors

The authors presented interesting research results. However, the work needs definite improvement and reconstruction.

The title necessarily needs to be changed. I disagree with the authors that they describe the agricultural soils of Jeju Island. This is not true, as they themselves write in lines 84 and 124-127  (repetition) citrus orchards in which they conducted their research are only 35% of all agricultural land! And what about the rest of the soils, such as vegetable crops with a very similar share, here no fertilizers and pesticides are used?

In the abstract, it should be stated at the beginning which heavy metals were investigated.

The formulation of anthropogenic agricultural activities is wrong, simply agricultural activities. Can they be without human contribution? The same is true for agricultural pesticides, why write agricultural?

In my opinion, the term single extraction is used incorrectly in the paper, after all, different bioavailability of heavy metals was determined in several extractions.

Introduction lines 52-52: Which heavy metals are involved? The problem with cadmium present in phosphates and phosphate fertilisers is well known. How does this relate to the results?

Line 61: change food web to food chain.

Lines 67-69: In that case, why was the total HM content in the study determined?

Lines 74-89: This is a description of the study area, please transfer to section 2.1. Remove repetition.

Lines 90-92: Are you sure? What area is affected? Globally, there are many such studies.

Line 97-98. I believe that the phrase “potential  polution sources of TP” is wrong. Why fertilise soils with phosphorus to pollute them?

Material… lines 112-114: What is the soil type? What is the unit? I suggest naming these soils after the WRB as well.

Lines 117-122. Why soil colour information? What does it add to the study?

Line 126: Food crops - what is it?

Figure 1: Etc- what it is?

Lines 142-147. The description of these methods should be in the manuscript and not in the supplementary materials. Please consider whether all the information is needed. Why was the pH determined in two solutions?

Line 149: Is aluminium (Al) classed as a heavy metal? There can be no mistake because the authors write seven HMs. This basically disqualifies the paper!

Lines 162-168:  This information is unnecessary. After all, we assume appropriate standards and determination procedures.

Line 171: Finally, how many of these heavy metals 6 or 7?

Lines 180-183: I do not agree with this choice of background. Where did the data come from? What assurance that the topsoil in the forests are not contaminated by HM, for example, from industrial activities. I suggest reading the paper by Kowalska et al. 2018 DOI:10.1007/s10653-018-0106-z

Lines 200-216: Why was the soil used here sifted through a <2mm sieve, and for the total HM content of the sieve <0.15mm. How can such results be compared?

Source origin… Lines 307-373: Some of the formulations are an over-interpretation of the results.

Conclusion from line 482 need to be reviewed.

Comments on the Quality of English Language

The manuscript definitely needs professional language proofreading. English is not my native language, but I see errors in sentence construction, for example in lines 18-20. Another good example of errors, are in the keywords phrase: source orgin - after all, they mean the same thing, and the question is but what?

Author Response

RESPONSES TO REVIEWER COMMENTS

Title: Accumulation and Origin of Phosphorus and Heavy Metals in Citrus Orchard Soils in Jeju Island, South Korea: Potential Ecological Risks and Bioavailability

First of all,
We would like to thank the Editor and Reviewers who reviewed the manuscript (Manuscript ID: water-2682077). We revised the manuscript based on the reviewer comments as follows:

Reviewer #1

Comment 1: 
The title necessarily needs to be changed. I disagree with the authors that they describe the agricultural soils of Jeju Island. This is not true, as they themselves write in lines 84 and 124-127 (repetition) citrus orchards in which they conducted their research are only 35% of all agricultural land! And what about the rest of the soils, such as vegetable crops with a very similar share, here no fertilizers and pesticides are used?
Response 
We agree with your advice.
As your advice, this manuscript was conducted only in citrus orchards rather than extensive agriculture. Thus, we have revised the title in “Revised Manuscript” as follows (including Supplementary material):

Accumulation and Origin of Phosphorus and Heavy Metals in Citrus Orchard Soils in Jeju Island, South Korea: Potential Ecological Risks and Bioavailability.

Comment 2: Abstract
In the abstract, it should be stated at the beginning which heavy metals were investigated.
Response 
We have added in the “Revised Manuscript” as follows (Lines 15-17):

This study investigated the accumulation of total phosphorus (TP) and heavy metals (HMs; Pb, Zn, Cu, Cd, Cr, and Ni) in citrus orchard soils in Jeju Island, Korea, identifying potential soil pollution sources using statistical analysis.

Comment 3: Abstract
The formulation of anthropogenic agricultural activities is wrong, simply agricultural activities. Can they be without human contribution? The same is true for agricultural pesticides, why write agricultural?
Response 
Thank you for your advice.
As advised, we have changed not only the abstract but also the entire “Revised Manuscript” as follows:

1. “anthropogenic agricultural activities” to “agriculture activities”
2. “agricultural pesticides” to “pesticides”

Comment 4: Abstract
In my opinion, the term single extraction is used incorrectly in the paper, after all, different bioavailability of heavy metals was determined in several extractions.
Response 
Thank you for your opinion.
Here, single extraction refers to a method of evaluating bioavailability using one extractant, unlike sequential extraction (using multiple extractants). In general, the term “single extraction” is used for these methods. However, “single extraction” used in this manuscript could cause confusion, so we have changed it all to “single-extraction”.

Comment 5: Introduction, Lines 52-52
Which heavy metals are involved? The problem with cadmium present in phosphates and phosphate fertilisers is well known. How does this relate to the results?
Response 
We have added the relevant heavy metals as follows (Lines 49-51):

Phosphorus fertilizers are regarded as a potential cause of HM (As, Cr, Zn, Cd, and Cu) accumulation in agricultural soils as they are produced from phosphate rock, which includes several naturally occurring HMs [4,10].

As you advise, phosphate rock and phosphorus fertilizers are well known to be closely related to cadmium. They are associated not only with cadmium, but also with phosphorus and other heavy metals. Our study area has unusual soil characteristics resulting in excessive application of phosphorus fertilizers. Thus, our study was conducted to understand the impacts of agricultural activities.

Comment 6: Introduction, Line 61
Line 61: change food web to food chain.
Response 
We have changed “food web” to “food chain” (Line 60).

Comment 7: Introduction, Lines 67-69
In that case, why was the total HM content in the study determined?
Response 
The total concentration of heavy metals is usually used as a simple indicator of the degree of contamination. In agricultural soils, assessments considering bioavailability rather than total concentration of heavy metals are more commonly used. As explained after the sentence you pointed out, heavy metals depend on various factors such as soil characteristics and the chemical fraction. Therefore, soluble extraction could provide an easier understanding of soil availability in soil than total metal concentrations.

Comment 8: Introduction, Lines 74-89
This is a description of the study area, please transfer to section 2.1. Remove repetition.
Response 
As you pointed out, we have decided to delete redundant sentences from the “Introduction” and “Materials and Methods (Section 2.1). However, we have decided to keep some sentences in the “Introduction” that are relevant to the purpose of the study. Then, we have deleted unnecessary citations and rearranged citation numbers in the “Revised Manuscript” (Lines 72-89).

Comment 9: Introduction, Lines 90-92
Are you sure? What area is affected? Globally, there are many such studies.
Response 
We agree with you.
We made a mistake when dividing the paragraphs. The sentence you pointed out is related to the previous paragraph. Here, we provide a limited description of Jeju Island in Korea. Thus, we have moved on to the previous paragraph (Lines 84-89).

Comment 10: Introduction, Lines 97-98
I believe that the phrase “potential polution sources of TP” is wrong. Why fertilise soils with phosphorus to pollute them?
Response 
We agree with your opinion.
Phosphorus is an essential nutrient for crop growth. We presented it at the beginning of the “Introduction”. However, phosphorus, which is over-fertilized in agricultural management, is known to be a potential pollution that causes eutrophication through movement of surface water and groundwater. Thus, we believe that there is no problem with the sentence you pointed out.

Comment 11: Material and Methods, Lines 112-114
What is the soil type? What is the unit? I suggest naming these soils after the WRB as well.
Response 
Thank you for your suggestion.
Herein, Jeju soils were classified according to the “Taxonomical Classification of Korean Soils”. Thus, we have decided to keep the sentence you pointed out.

Comment 12: Lines 117-122
Why soil colour information? What does it add to the study?
Response 
Thank you.
We found that the detailed description of Jeju soil characteristics actually caused confusion in the manuscript. Therefore, we have decided to delete the detailed description of Jeju soil characteristics that were less relevant to the purpose of the manuscript. Also, we have changed Figure 1 in the “Revised Manuscript”.

Comment 13: Line 126
Food crops - what is it?
Response 
Thank you.
We have decided to delete the sentences in “Section 2.1” because they were redundant with sentences presented in the “Introduction” (Lines 79-80). We believe that deleting the sentences in “Section 2.1” would have no impact on the “Revised Manuscript”.

Comment 14: Figure 1
Etc- what it is?
Response 
We are confused by the detailed Jeju soil characteristics in the manuscript. As suggested in “Comment 12”, we have changed “Figure 1” to avoid confusion. Thus, we have solved what you pointed out.

Comment 15: Lines 142-147
The description of these methods should be in the manuscript and not in the supplementary materials. Please consider whether all the information is needed. Why was the pH determined in two solutions?
Response 
Thank you for your advice.
Soil physicochemical analysis was presented because it need to be performed basically to understand the soil characteristics. However, this manuscript contains most of the pollution-related considerations. Therefore, considering the importance of the manuscript, the detailed analysis method was excluded from this manuscript and presented in the “Supplementary Materials”. Also, pH (NaF), one of the soil pH, is a method to evaluate the characteristics of volcanic ash soil, which is a unique Jeju soil. Information on the characteristics of Jeju soils, which is volcanic ash soils, are presented in “Section 2.2”. Thus, we have decided to keep it (Lines 124-129).

Comment 16: Line 149
Is aluminium (Al) classed as a heavy metal? There can be no mistake because the authors write seven HMs. This basically disqualifies the paper!
Response 
Aluminum (Al) is a constituent of the Earth’s crust and is a light metal rather than a heavy metal. 
In this manuscript, Al caused confusion as a background concentration item required for the assessment of heavy metal contamination. Thus, we have revised them all as follows (Lines 131-133, 216-218, 241-243, 261-262):

Lines 131-133: The concentrations of TP and six HMs (Pb, Zn, Cu, Cd, Cr, and Ni) were analyzed using aqua regia (HNO3:HCl = 1:3) digestion through the following pretreatment processes (including Al of the Earth’s crust) [55].

Lines 216-218: The concentration distributions of TP and six HMs (Pb, Zn, Cu, Cd, Cr, and Ni) in soils (topsoil and subsoil) collected from citrus orchards and forestlands are shown as a box-and-whisker plot (including Al of the Earth’s crust) (Figure 2).

Lines 241-243; Figure. 2. Box and whisker plots showing the distribution of total phosphorus (TP) and heavy metals (Pb, Zn, Cu, Cd, Cr, and Ni) in topsoil (Top) and subsoil (Sub) collected from citrus orchards (OS) and forestland (FS) in Jeju Island, South Korea (including Al of the Earth’s crust). 

Lines 261-262: Figure 3. Spatial distributions of TP and heavy metals (Pb, Zn, Cu, Cd, Cr, and Ni) in topsoil collected from citrus orchards in Jeju Island, South Korea (including Al of the Earth’s crust).

Comment 17: Lines 162-168
This information is unnecessary. After all, we assume appropriate standards and determination procedures.
Response 
As your opinion, we performed according to appropriate standards and analytical procedures. Nevertheless, we believe that it is necessary to present quality control data for reliability verification of the analysis results. Thus, we have decided to keep what you pointed out (Lines 145-152).

Comment 18: Line 171
Finally, how many of these heavy metals 6 or 7?
Response 
There is no problem with the written sentence (Line 154).
Previously, it was suggested as seven heavy metals, causing confusion
Al is a background concentration item used to assess heavy metal contamination.

Comment 19: Lines 180-183
I do not agree with this choice of background. Where did the data come from? What assurance that the topsoil in the forests are not contaminated by HM, for example, from industrial activities. I suggest reading the paper by Kowalska et al. 2018 DOI:10.1007/s10653-018-0106-z
Response 
Thank you for your advice.
We also appreciate your suggestions for citation to improve the quality of our manuscript. We selected forestland area with low anthropogenic impacts from industrial activities because it was appropriate as a control area. Above all, we believe that since there are no cities and industrial complexes around the forestland, there would have been no anthropogenic heavy metal pollution impacts. Jeju soils, which is characterized by volcanic ash, is very different in soil type from that of other regions, and some heavy metals are known to have high background concentrations due to this characteristic. In your proposed citation, geochemical background concentrations vary depending on regional differences and soil types. Thus, we believe that the background concentration used in our study will not be a problem for manuscript quality (Lines 162-165).

Comment 20: Lines 200-216
Why was the soil used here sifted through a <2mm sieve, and for the total HM content of the sieve <0.15mm. How can such results be compared?
Response 
Thank you for your opinion.
Here, we followed standard analytical methods for soil pollution and most widely used procedures in citation. Thus, we believe there no problem with our manuscript results. For single-extraction, the soil samples, unlike total HMs, were intended to evaluate the chemical fraction adsorbed to soil particles, resulting in the use of raw materials of less than 2 mm (Line 187). In addition, we believe that comparative evaluation would be possible because the amount of samples used in the single-extraction was much larger than the total HM.

Comment 21: Lines 307-373
Source origin: Some of the formulations are an over-interpretation of the results.
Response 
Thank you.
Here, we used CA and PCA statistical analysis techniques, which are commonly used, to evaluate the origin of pollution. Thus, we tried to interpret out results through sufficient consideration using CA and PCA techniques. We believe that sufficient consideration would have helped to improve the quality of the manuscript (Lines 289-348).

Comment 22: Conclusion, Line 482
Conclusion from line 482 need to be reviewed.
Response 
We have rewritten what you suggested as follows (Lines 461-464):

Furthermore, the Igeo and EF assessments suggested that Zn, Cu, and Cr in citrus orchard topsoil were affected by anthropogenic sources; in particular, Cu showed relatively higher contamination levels and enrichment than other HMs.

Comment 23: Lines 18-20
The manuscript definitely needs professional language proofreading. English is not my native language, but I see errors in sentence construction, for example in lines 18-20. Another good example of errors, are in the keywords phrase: source orgin - after all, they mean the same thing, and the question is but what?
Response 
We have completed professional English proofreading, including the sentences you pointed out as follows (Abstract and Keywords):

Abstract (Lines 19-20): TP, Zn, Cu, and Cr concentrations in citrus orchard topsoil were significantly higher than those in forestland soils, indicating their accumulation in the surface layer.

Keywords (Lines 30-31): citrus orchard; phosphorus; heavy metals; accumulation; origin; single extraction; bioavailability; correlation analysis; principal component analysis

Thus, we believe that there no problems with the English language in the “Revised Manuscript”.

Reviewer 2 Report

Comments and Suggestions for Authors

Dear authors,

Many congratulations!

In my opinion your manuscript is ready to be accept.

Author Response

Response 
Thank you for reviewing our manuscript.

Reviewer 3 Report

Comments and Suggestions for Authors

The paper “Accumulation and Origin of Phosphorus and Heavy Metals in Agricultural Soils in Jeju Island, South Korea: Potential Ecological Risks and Bioavailability” covers an important topic related to the bioavailability of HMs and phosphorus present in agricultural soils due to anthropogenic and natural influences. The paper is easy to read, well-structured and motivated. Sample collection, sample preparation and analytical methods used are relevant, as well as references to the literature.

Minor revision needed. Please clarify the following questions:

2.3. Phosphorus and heavy metal analysis

It is stated that seven HMs have been identified, including Al, but no quality control data for Al is provided. There is also no quality control data for P.

2.4. Pollution assessment of heavy metals

The control soil samples belong mostly to brown forest and black soil groups, while the citrus orchards are located mainly on Very Dark Brown soil. Please explain why you chose this type of control. How correct is the use of such control? Does the correction constant for crustal origin effects cover this discordance, too? Please 

EF is not a method, but a parameter.

Figure 3 shows not only spatial distributions of TP and HM concentrations, but also the altitude of collection points. Is this important for data interpretation since the correlations between elemental content and altitude is not reported?

Author Response

RESPONSES TO REVIEWER COMMENTS

Title: Accumulation and Origin of Phosphorus and Heavy Metals in Citrus Orchard Soils in Jeju Island, South Korea: Potential Ecological Risks and Bioavailability

First of all,
We would like to thank the Editor and Reviewers who reviewed the manuscript (Manuscript ID: water-2682077). We revised the manuscript based on the reviewer comments as follows:

Reviewer #2

Comment 1: 2.3. Phosphorus and heavy metal analysis
It is stated that seven HMs have been identified, including Al, but no quality control data for Al is provided. There is also no quality control data for P.
Response 
We appreciate your comments.
In fact, we were unable to obtain certified reference materials containing Al and P at the time of the study. Therefore, we could not present reliability results for Al and P. However, we have completed sufficient verification through an analytical quality control with standard solutions of Al and P. Thus, we believe that the concentrations of Al and P in the soils presented in the manuscript are not problematic.

Comment 2: 2.4. Pollution assessment of heavy metals
The control soil samples belong mostly to brown forest and black soil groups, while the citrus orchards are located mainly on Very Dark Brown soil. Please explain why you chose this type of control. How correct is the use of such control? Does the correction constant for crustal origin effects cover this discordance, too? Please 
Response 
Thank you for your opinion.
We selected forestland, which is less related to the anthropogenic impact from industrial activities, because it was appropriate as a control sites. However, we found that the detailed description of Jeju soil characteristics actually caused confusion in the manuscript. Therefore, we have decided to delete the detailed description of Jeju soil characteristics that were less relevant to the purpose of the manuscript. Also, we have changed Figure 1 in the “Revised Manuscript”.

Comment 3: 2.4. Pollution assessment of heavy metals
EF is not a method, but a parameter.
Response 
We have changed from “method” to “parameter” in the “revised manuscript” (Line 170).

Comment 4: Figure 3
Figure 3 shows not only spatial distributions of TP and HM concentrations, but also the altitude of collection points. Is this important for data interpretation since the correlations between elemental content and altitude is not reported?

Response 
We appreciate your advice.
The distribution of TP and HMs concentrations has no correlation with altitude.
Here, we only expressed the explanation of the Jeju sampling sites to make it easier to understand, but it caused confusion. Thus, we have decided to remove only the legend for altitude from Figure 3 in “Revised Manuscript” to avoid confusion.

Round 2

Reviewer 1 Report

Comments and Suggestions for Authors

In my opinion, the number of keywords is too high, e.g. single extraction; bioavailability or concerning statistics.

I do not fully agree with the authors on the choice of geochemical background, but accept their explanation.

Author Response

Response 
Thank you for your opinion.
We have deleted keywords related to statistics as follows (Lines 30-31):

citrus orchard; phosphorus; heavy metals; accumulation; origin; single extraction; bioavailability
